# Distribution and Neurochemical Characterization of Dorsal Root Ganglia (DRG) Neurons Containing Phoenixin (PNX) and Supplying the Porcine Uterine Cervix

**DOI:** 10.3390/cells14231847

**Published:** 2025-11-23

**Authors:** Urszula Mazur, Paulina Kuśmierek, Paweł Janikiewicz, Mariusz Krzysztof Majewski, Agnieszka Bossowska

**Affiliations:** Department of Human Physiology and Pathophysiology, University of Warmia and Mazury in Olsztyn, Warszawska 30, 10-082 Olsztyn, Poland; urszula.mazur@uwm.edu.pl (U.M.); paulina.kusmierek@uwm.edu.pl (P.K.); pawel.janikiewicz@uwm.edu.pl (P.J.); agnieszka.bossowska@uwm.edu.pl (A.B.)

**Keywords:** phoenixin-14, pig, dorsal root ganglia, uterine cervix, neuropeptides, immunofluorescence

## Abstract

One of most important parts of the female genital tract is the uterine cervix, both from the anatomical as well as physiological points of view. As there is currently a lack of detailed information on the presence, distribution pattern(s), and the chemical coding of phoenixin (PNX)-containing dorsal root ganglia (DRG) neurons supplying the porcine uterine cervix, this study, using combined retrograde tracing and double-immunofluorescence techniques, was aimed at analyzing (i) the distribution pattern of uterine cervix-supplying sensory neurons (UC-SNs) at the particular spinal cord levels, (ii) their intraganglionic distribution, and (iii) the patterns of PNX co-expression with other biologically active substances. UC-SNs were identified by the presence of deposits of Fast Blue (FB), in DRG of thoracic (Th10–Th15), lumbar (L1–L5) and sacral (S2–S4) spinal cord segments. FB^+^/PNX^+^ neurons constitute approximately 33% of all UC-SNs, 73% at the L, and 27% at the S neuromeres. These neurons were mainly small sized (52%), with a slightly smaller population of medium-sized cells (40%), while large-diameter cells made up the least numerous population (8%). The vast majority of FB^+^/PNX^+^ neurons simultaneously contained calcitonin gene-related peptide (CGRP; 80.9%) or substance P (SP; 77.9%); one-third of them showed immunoreactivity toward neuronal nitric oxide synthase (nNOS; 34%), while PNX^+^ UC-SNs containing pituitary adenylate cyclase-activating polypeptide (PACAP), galanin (GAL), calretinin (CRT), or somatostatin (SOM) formed significantly smaller populations (21.4%, 7.4%, 3.1%, and 0.7%, respectively). The results of the present study demonstrate the presence of PNX in DRG UC-SNs, and its co-occurrence with numerous neurotransmitters suggesting a putative role for this neuropeptide in the transmission of various types of sensory information and possible effects on the functioning of this organ in the body.

## 1. Introduction

Phoenixin-14 (PNX) is a bioactive peptide discovered in the rat brain [1]. It is highly conserved across mammalian and non-mammalian species with PNX-14 and -20 being the most prominent bioactive forms giving rise to its physiological importance [2]. PNX-14 is identical among multiple species including human, rat, mouse, pig, and dog whereas PNX-20 differs in one amino acid between the coding region of human, canine, or porcine sequences. Despite these differences in sequence length, they appear to function similarly [2]. Both are evolutionarily conserved products of post-translational processing of the prohormone SMIM20 with the difference that PNX-20 is expressed mainly in the hypothalamus, whereas PNX-14 is more abundant in the heart and the spinal cord [3,4].

Simultaneously with the discovery of PNX in 2013, Lyu and Yosten proposed two “main” functions for PNX: the participation in sensory transmission, as well as the control of reproductive processes [1,2]. Recent literature reports indicate that PNX plays a significant role in the activity of the hypothalamic–pituitary–gonadal (HPG) axis: it has a direct effect on the secretion patterns of pituitary gonadotropins [5], most probably by raising the production of gonadoliberin (GnRH) receptor mRNA, controlling in this way the release of luteinizing hormone (LH) [2]. Furthermore, the presence of PNX in neurons of magnocellular paraventricular and supraoptic nuclei of the hypothalamus suggests the secretion of this peptide into the systemic circulation and therefore its possible transport to sites of action in the periphery [5]. To date, research has been conducted on the possible functions of PNX in the ovary [3], including isolated ovarian follicles [6] and the uterine endometrium. In vitro studies on human ovarian follicles have shown that PNX, through activation of the G-protein-coupled receptor 173 (GPR173), enhances the activity of genes related to the ovarian follicle development, promoting their growth by accelerating the proliferation of granulosa cells, which leads to an increased number of ovulated oocytes [6,7]. GPR173 levels increased in ovarian follicles as estrus progressed [7]. Later studies have shown that PNX and its receptor are produced locally in the porcine corpus luteum and participate in the regulation of the follicular phase [6] and luteal phase, and their levels change during the estrous cycle [8]. There is evidence that the effect of PNX on the secretion of reproductive hormones is due to its interaction with other peptides involved in reproduction, such as nesfatin-1, which can potentiate the effect of PNX on the release of reproductive hormones such as LH, follicle-stimulating hormone (FSH), and testosterone in male rats [9]. Elevated levels of PNX have been observed in the blood of women suffering from the polycystic ovary syndrome (PCOS) [10], in both lean and obese patients [11] and in girls with precocious puberty, indicating its possible role in puberty [12]. In turn, reduced PNX levels were observed in serum in patients with endometriosis, which may suggest its involvement in the pathogenesis of the disease, as well as in the exacerbation of pelvic pain associated with cyclical changes in the endometrium [13]. Further studies have shown that endometrial cells increase the synthesis of the SMIM20 precursor in response to low PNX levels, which may suggest a potential role of PNX in improving epithelial dysfunction [14] and may be an effective strategy for using PNX in the treatment of endometriosis.

PNX was also detected in sites traditionally not directly associated with reproduction. The first studies in rodents shed light on the sensory nature of this peptide, through its presence in the superficial layers of the dorsal horn (DH) of the spinal cord, in a fairly numerous subpopulation of dorsal root ganglia (DRG) neurons and in the trigeminal and nodose ganglion cells [1]. In the domestic pig the presence of PNX has been reported in nerve fibers forming a dense plexus in laminae I and II of the DH [15]. This topographic distribution of PNX^+^ terminals indicates that these nerve fibers originated mainly from DRG neurons [16].

Although increasing evidence suggests that PNX participates in the controlling of numerous gonadal and endometrial functions, data are still lacking regarding both the presence and origin of PNX^+^ nerve fibers in the cervix. Because our pilot study (unpublished data) observed numerous PNX^+^ fibers in the cervical wall, the location of which could suggest their afferent nature, in this study we decided to investigate both the sources of PNX^+^ innervation and the chemical coding of sensory cells supplying the uterine cervix using a combination of retrograde neuronal labeling techniques and the double immunofluorescence staining. Furthermore, in a previous experiment [16], the chemical coding of PNX^+^ neurons at all levels of the spinal cord was initially characterized, providing a solid basis for analyzing both the distribution and the neurochemical characteristics of DRG cells supplying the uterine cervix. It was decided to conduct this research on domestic pigs because they share many embryonic, anatomical, and physiological similarities with humans and therefore appear to be a suitable animal model for biomedical research, including those focusing on the reproductive tract [17].

## 2. Materials and Methods

### 2.1. Experimental Animals

The research was carried out on sexually immature female pigs of the Large White Polish race (*n* = 6; age: 8–12 weeks; body weight (b.w.): 15–20 kg). All animals from the industrial fattening farm, after being delivered to the animal house of the Faculty of Veterinary Medicine, University of Warmia and Mazury in Olsztyn, were kept in a standard regime for this type of laboratory animals, with unlimited access to drinking water and appropriately portioned, standard feed mixture appropriate for their age (Grower Plus, Wipasz, Wadag, Poland). The animals were housed and treated in accordance with the rules of the both during the animal housing (pre- and postoperative periods) and during planned surgical procedures; the guidelines contained in the decision (No. 40/2020 from 22 July 2020) of the Local Ethics Committee for Animal Experimentation in Olsztyn (affiliated with the National Ethics Commission for Animal Experimentation, Polish Ministry of Science and Higher Education) were strictly followed to both minimize the number of animals used in this experiment and minimize their suffering.

### 2.2. Anesthesia and Surgical Procedures

Each planned surgical procedure was preceded by the induction of full anesthesia. For this purpose, each animal received, 30 min before the administration of the main anesthetic (Thiopental; (sodium pentobarbital), Sandoz, Warsaw, Poland; 0.5 g per animal, administered intravenously), subcutaneous injections of atropine (Polfa, Warsaw, Poland, 0.05 mg/kg b.w.) and intramuscular injections of azaperone (Stresnil, Janssen Pharmaceutica, Beerse, Belgium, 2.5 mg/kg b.w.). The corneal test was used each time as a measure of the depth of anesthesia achieved in each individual animal. The uterine cervix was carefully exposed after a midline laparotomy, and a total volume of 40 µL of 5% aqueous solution of the fluorescent retrograde tracer Fast Blue (FB) (Dr. K. Illing KG & Co GmbH, Gross Umstadt, Germany) was injected into its wall. The tracer was administered in multiple injections (1 µL of the dye solution per injection using a Hamilton microsyringe equipped with a 26S gauge needle) under the serosa along its entire length, maintaining a similar distance between the injection sites. Possible leakage of FB from the injection channel and, consequently, contamination of the surrounding tissues, which would have been the reason for excluding the animal from the study group, was prevented by keeping the needle in the injection channel for at least one minute without moving it. The surface of the injected cervix was then gently wiped with gauze moistened with physiological saline after each injection. Three weeks later, which is an optimal time for the retrograde tracer to be transported to the DRG from the cervix, all the pigs were deeply anesthetized (following the same procedure as described above) and after the cessation of breathing, transcardially perfused with freshly prepared 4% paraformaldehyde in 0.1 M phosphate buffer (pH 7.4). Afterwards, the spinal cords with DRG were collected from all the animals. As previously described in detail [16], individual spinal cord levels with corresponding DRGs were postfixed, thoroughly washed in 0.1 M phosphate buffer, and finally transferred to and stored in 18% buffered sucrose at 4 °C until sectioning.

### 2.3. Sectioning the Tissue Samples

Based on the previously presented detailed protocol for handling the examined tissues [16], bilateral DRG from thoracic (Th: Th10–Th15), lumbar (L: L1–L6), and sacral (S: S1–S4) spinal cord levels were pairwise located on a frozen block of the optimal cutting temperature (OCT) compound in a manner allowing easy determination of the position of individual subdomains of DRG sections under the microscope. Frozen DRG pairs were cut with an HM525 Zeiss freezing microtome on transverse 10 µm-thick serial sections (four sections on each slide) and mounted on chrome alum-gelatine-coated slides.

### 2.4. Estimation of the Total Number of FB^+^ DRG Uterine Cervix-Supplying Neurons (UC-SNs)

To calculate the relative number of UC-SNs FB^+^ nerve cells the previously used protocol [18] was applied. Briefly, retrogradely labeled UC-SNs were counted in every fourth section (to avoid double-counting of the same neuron) of both the right and the left DRG of all the animals. It should be stressed that only neurons with a clearly visible nucleus were counted. The results were pooled for every experimental animal and statistically analyzed, and the mean number of FB-positive cells was calculated. The total number of FB^+^ nerve cells counted in all collected DRG from a particular animal as well as the relative frequencies of perikarya in the ganglia belonging to the individual neuronal size classes were presented as mean ± SEM (standard error of mean). The diameter of the FB^+^ perikarya was measured by means of an image analysis software (version 3.02, Soft Imaging System, Münster, Germany), and data were used to divide UC-SNs into the three size-classes: small (average diameter up to 40 µm), medium-sized (diameter 41–70 µm), and large afferent cells (diameter > 71 µm), while the statistical analysis was performed with Graph-Pad Prism 8 software (GraphPad Software, La Jolla, CA, USA).

### 2.5. Immunohistochemical Procedures

Serial 10 µm-thick DRG sections were processed for double-labeling immunofluorescence (following the details of an earlier described protocol [19] using the combinations of primary immunosera raised in different species). After the immersion of tissues in a blocking solution containing 0.1% bovine serum albumin, 1% Triton X 100, 0.01% sodium azide (NaN_3_), 0.05% thimerosal, and 10% normal horse serum in 0.01 M phosphate-buffered saline (PBS) for 1h at room temperature to reduce non-specific background staining, sections were repeatedly rinsed in PBS, and then incubated overnight at room temperature with antisera against calcitonin gene related peptide (CGRP), calretinin (CRT), galanin (GAL), neuronal nitric oxide synthase (nNOS), pituitary adenylate cyclase-activating polypeptide (PACAP), PNX, somatostatin (SOM), and substance P (SP).

Primary antisera were, as described previously [16], visualized by the use of either fluorescein isothiocyanate (FITC)-conjugated rat-, guinea pig-, or mouse-immunoglobulin G (IgG) specific secondary antisera or streptavidin (CY3)-cojugated rabbit-specific antiserum. After rinsing the tissue with PBS, Fluoromount™ Aqueous Mounting Medium (Sigma Aldrich, Saint Louis, MO, USA, F4680) was applied to the section and then the tissue was covered with a coverslip. The colocalization of PNX with other biologically active substances in FB^+^ neurons was examined on the consecutive sections obtained from all DRGs studied. Immunostained FB^+^ DRG perikarya were evaluated under an Olympus BX61 microscope (Olympus, Hamburg, Germany) equipped with epifluorescence filter for FB and an appropriate filter set for CY3 or FITC. Finally, the obtained data were pooled in all the animals and presented as mean ± SEM, referring to the number of animals, and statistically analyzed by Student’s two-tailed *t*-test for unpaired data. The results were statistically analyzed using Graph-Pad Prism 8 software (GraphPad Software, La Jolla, CA, USA). The differences were considered to be significant at *p* < 0.05. List of primary antisera and secondary reagents used in the study—see Appendix A.

### 2.6. Control of Specificity of Immunohistochemical Procedures

In order to confirm both the lack of tissue contamination with tracer and the specificity of the antibodies used, two previously used protocols [20] were applied in this experiment: (i) Thorough macroscopic examinations of the sites of FB injections and the tissues adjacent to the cervix were performed before collecting the ganglia. The sites of the injections were easily identified by the yellowish-labeled deposition left by the tracer within the uterine cervix wall. Moreover, the injection sites were also observed in the UV lamp rays in the dark room. The tissues adjacent to the cervix were not found to be contaminated with the tracer. All these procedures excluded any leakage of the tracer and validated the specificity of the tracing protocol. (ii) Standard controls, i.e., preabsorption for the neuropeptide antisera (20 μg of appropriate antigen per 1 mL of the corresponding antibody at working dilution; all antigens purchased from Peninsula, R&D Systems, SWANT, Sigma, or Phoenix), as well as omission and replacement of the respective primary antiserum with the corresponding non-immune serum completely abolished immunofluorescence and eliminated specific immunolabelling. List of antigens used in pre-absorption tests—see Appendix A.

## 3. Results

### 3.1. The Distribution Pattern and Morphometrical Characteristics of Porcine UC-SNs

#### 3.1.1. Distribution Pattern of FB^+^ Neurons

After the administration of FB into the uterine cervix wall, FB^+^ neurons were found in both the left and the right DRG of all the pigs studied. The total number of sensory neurons innervating the uterine cervix per animal ranged from 621 to 735 retrogradely labeled perikarya (682.7 ± 33.2; mean ± SEM). The number of FB^+^ neurons per animal ranged from 320 to 418 retrogradely labeled neurons (358.7 ± 30.1) in the left DRG and from 289 to 355 FB-positive sensory nerve cells (320 ± 18.8) in the right ones. No statistically significant differences in the number of FB-containing sensory neurons were observed between the left and the right ganglia studied.

FB-positive neurons were observed in DRG collected from all levels of the spinal cord studied. Approximately 64% (64.1 ± 1.5%) of all FB^+^ sensory neurons were located in DRG of S level (S2–S4) of spinal cord, less than 36% (35.3 ± 1.4%) of FB^+^ nerve cells were observed in the L DRG (L1–L5), while only single retrogradely traced sensory neurons were found in thoracic DRG (Th12–Th15; due to the fact that FB^+^ cells observed in the DRGs of this spinal cord level constituted only 0.6 ± 0.6% of the total UC-SNs population, it was decided not to evaluate their chemical coding patterns in the remainder of the experiment). Details concerning the relative frequency of FB^+^ sensory neurons innervating the uterine cervix and observed in the left and the right L and S DRG are shown in Figure 1 and Table 1.

As mentioned above, because only a few nerve cells were presented at the Th level, we focused our attention mainly on retrogradely labeled neurons forming two distinct “centers” located bilaterally in the DRG of the L1–L5 (“L–center”) as well as S2–S4 (“S–center”) levels of the spinal cord. There were statistically significant differences in the occurrence of FB^+^ cells between these two major L and S “centers” (*p* = 0.0002) (Figure 1). The UC-SNs were absent in the bilateral L6 and S1 DRG.

#### 3.1.2. Morphometrical Characteristics of FB^+^ Sensory Neurons

FB^+^ DRG neurons innervating the uterine cervix belonged to the following three cell-sized classes: small (neurons with an average diameter of up to 40 µm), medium (neurons with a diameter ranging from 41 µm to 70 µm), and large (neurons with a diameter of up to 71 µm). In general, almost half of FB^+^ sensory neurons belonged to the small perikarya (48.7 ± 0.5%), which slightly predominated over the medium-sized neurons (40.9 ± 1.0%), whereas large cells (10.4 ± 0.7%) formed the smallest population of all DRG FB^+^ cells examined. There were no statistically significant differences in the number of FB^+^ sensory neurons belonging to these three size classes between the left and the right DRG.

Considering the two main “centers” of FB^+^ cells innervating the uterine cervix, in DRG located at the L level of spinal cord, the occurrence of mainly medium-sized cells was observed (45.5 ± 3.3%), while a slightly smaller population was formed by small cells (34.2 ± 5.4%); large-diameter neurons constituted approximately one fifth of the cells counted (20.3 ± 2.4%). In comparison, the results obtained from the DRG of the S level of spinal cord were different: the largest population was composed of small-sized (56.9 ± 3.8%) cells, with almost a half as many medium-sized cells (37.8 ± 3.1%), while the large neurons (5.2 ± 0.8%) were only occasionally found in all the DRGs studied from this level.

On the one hand, no statistically significant differences were observed in the number of small, medium, and large diameter cells in the subpopulations of traced neurons in the left- and the right-sided DRGs studied; on the other hand, such statistically significant differences were observed in the number of small and large diameter cells in the L and S “centers”. Details regarding the relative frequency of FB^+^ sensory neurons belonging to the different size-classes, as well as the differences observed in their intraganglionic distribution were presented in Table 2. Representative images of FB^+^ DRG neurons belonging to the mentioned size classes are presented in Figure 2.

#### 3.1.3. Intraganglionic Distribution Patterns of FB^+^ Neurons

To determine the intraganglionic distribution patterns of FB^+^ sensory neurons in all the DRG examined, an identical “mask” (as used in our previous study [16], shown in (Figure 3), was applied to each of analyzed ganglionic sections.

FB^+^ neurons were present in all five ganglion domains: central (Cn), caudal (Cd), cranial (Cr), peripheral (P), and middle (Md). In general, the largest number of FB^+^ nerve cells were present in the Cr domain (35.1 ± 2.2%). Slightly fewer neurons were observed in the Cd and P domains (26.0 ± 1.4% and 19.1 ± 0.6%, respectively). The smallest population of FB-positive neurons were those located in the Md (12.1 ± 0.8%) and Cn domains (7.7 ± 0.8%) of all the DRGs studied.

When focusing on the DRG of L level of the spinal cord, a marked accumulation of retrogradely labeled FB^+^ nerve cells was found in the Cd domain (35.0 ± 1.1%). Slightly fewer cells were observed in the Cr and P domains (22.6 ± 0.6% and 19.4 ± 1.2%, respectively), while the UC-SNs were the least numerous in the Cn and Md domains (10.7% ± 1.5% and 12.3 ± 1.9%, respectively). In the case of DRG located at the S level of the spinal cord, the largest population of FB^+^ sensory neurons was observed in the Cr subdomain (42.2 ± 3.0%), while a slightly smaller number of retrogradely labeled nerve cells has been present in the Cd and P regions (21.2 ± 1.7% and 19.0 ± 0.7%, respectively). Similarly to the L level, the smallest number of UC-SNs was found in the Md (11.6 ± 2.7%) and Cn (6.0 ± 0.4%) regions of DRG of S level of the spinal cord. Details regarding the intraganglionic distribution pattern of retrogradely labeled sensory neurons in the left and the right DRG examined are presented in Table 3.

### 3.2. Distribution Pattern and Morphometrical Characteristics of Porcine FB^+^/PNX^+^ UC-SNs

The collected data indicate that there are differences in the number of FB^+^ nerve cells containing any of the biologically active substances tested between the L and the S “centers” of DRG. Taking into account all the DRGs supplying the uterine cervix, 78.6 ± 2.6% of them contained at least one of the substances tested. However, when analyzing the expression patterns of the compounds tested in the L and the S “centers” separately, neurons expressing at least one of the substances were studied; while as many as 98.2 ± 0.8% of retrogradely labeled L DRG neurons contained at least one of the tested agents, only 65.0 ± 2.6% of traced sensory cells in the S ganglia contained at least one of the tested substances. The type of transmitter(s) used by the “immunonegative” neurons found in this study remains to be elucidated.

#### 3.2.1. Distribution Pattern of FB^+^/PNX^+^ Neurons

PNX^+^ UC-SNs were observed bilaterally in all the DRG examined. The total number of FB^+^/PNX^+^ sensory neurons *per* animal ranged from 175 to 265 (194 ± 36.3), which represented 33.3 ± 1.7% of all FB-positive sensory nerve cells counted in the DRG examined, of which in the left ganglia, the number of FB^+^/PNX^+^ neurons *per* animal ranged from 95 to 149 (127.7± 16.5), whereas in the right ganglia it ranged from 80 to 129 cells (108.3 ± 14.6). No statistically significant differences in the number of FB^+^/PNX^+^ containing sensory nerve cells were observed between the left and the right ganglia. However, when comparing the relative frequencies of PNX^+^ neurons in DRG of L vs. S “centers”, the vast majority of these cells were present in L DRG (73.4 ± 3.8%), whereas in S ganglia these perikarya were observed in much lower number (26.6 ± 2.5%).

#### 3.2.2. Morphometrical Characteristics of FB^+^/PNX^+^ DRG Neurons

While generally slightly more than a half of the UC-SNs belonged to the small-diameter class of neurons (52.0 ± 1.1%), 40.5 ± 0.7% of the retrogradely labeled PNX^+^ nerve cells were medium sized; large cells constituted less than 10% of the population examined (7.5 ± 1.9%). However, when the pattern of the L and the S DRG belonging to the three size classes was analyzed separately, some differences between these two anatomical levels were observed.

In the L DRG, the most numerous classes of FB^+^/PNX^+^ cells were medium-sized neurons (47.0 ± 1.7%); small-diameter perikarya constituted a slightly less numerous group (44.6 ± 44.6 ± 0.6%); large cells were significantly less numerous (8.3 ± 1.8%). In contrast to the pattern observed in the L DRG, in the S DRG, small-diameter PNX^+^ cells were most numerous (72.4 ± 2.5%), while medium-sized neurons constituted only 22.2 ± 2.1%. Similarly to the L ganglia, large diameter cells constitute a significant minority in the S DRG (5.3 ± 1.4%).

It should be emphasized that while no statistically significant differences in the distribution of the cells examined were observed between the left- and the right-sided DRG of either the L or the S spinal cord levels, such differences in the number of small- and medium-sized FB^+^/PNX^+^ cells were found between the L and the S “centers” of the sensory innervation of the uterine cervix. Details concerning the relative frequency of FB^+^/PNX^+^ sensory neurons of the different size classes in left and the right DRG of both anatomical levels, as well as statistically significant differences observed are shown in Table 4 and Figure 4.

#### 3.2.3. Intraganglionic Distribution Patterns of FB^+^/PNX^+^ Neurons

In general, when summing all FB^+^/PNX^+^ cells found in the DRG examined, regardless of their location in the L or the S spinal cord levels, a slightly greater accumulation of retrogradely labeled neurons was observed in the Cd region (29.5 ± 1.7%) of the ganglia, while 21.7 ± 0.3% and 20.6 ± 1.3% of them were located in the P and Cr domains, respectively. In the remaining subdomains of the ganglion (Md and Cn), 14.2 ± 2.5% and 14.0 ± 2.8% of all retrogradely labeled PNX^+^ cells were found. Also, in terms of the intraganglionic distribution of FB^+^/PNX^+^ cells, differences were observed between the DRG of L and S levels: while at both levels of the spinal cord FB^+^/PNX^+^ cells constituted similarly numerous subpopulations in the Cd domain (33.2 ± 2.0% vs. 33.4 ± 0.5% of the examined neurons), in the Cr domain they were almost twice as numerous as in the DRG of S levels (17.2 ± 0.8% vs. 33.4 ± 0.5%). While in the P domain the numbers of cells examined were similar in the DRG of both spinal cord levels examined (L: 20.4 ± 3.4% vs. S: 19.2 ± 1.8%), in both the Md domain and the Cn domain FB^+^/PNX^+^ perikaryons were slightly more numerous in the L DRG (L: Md: 14.6 ± 1.8% vs. S: 9.5 ± 1.2% and L: Cn: 14.6 ± 1.8% vs. S: 5.5 ± 0.5%, respectively). The details of the intraganglionic distribution pattern of retrogradely labeled PNX^+^ neurons in the individual domains of the left and the right DRG examined taken from the L and S levels of the spinal cord, as well as statistically significant differences, are presented in Table 5.

### 3.3. Immunohistochemical Characteristic of PNX-Containing UC-SNs in DRG Studied

Analysis of consecutive, double-immunohistochemically labeled serial sections of DRG studied has revealed that the vast majority (97.3 ± 1.4%) of all FB^+^/PNX^+^ cells showed simultaneous co-expression of the other substances tested, although the number of cells in the individual subpopulations differed from each other. The remaining FB^+^/PNX^+^ perikarya (2.7 ± 1.4%) were immunopositive only for PNX.

#### 3.3.1. Characteristics of FB^+^ Neurons Co-Localizing PNX and CGRP

The most numerous subset of FB^+^/PNX^+^ sensory neurons was composed of cells simultaneously containing CGRP (Figure 5(A1–4)). These cells accounted for 80.9 ± 2.3% of the total population of FB-/PNX-positive neurons and were observed in both the left- and the right-sided DRG (80.0 ± 1.4% and 83.3 ± 4.9%, respectively). While FB^+^/PNX^+^/CGRP^+^ neurons were uniformly distributed in DRG from both sides of the spinal cord, statistically significant differences in the numbers of such coded cells were observed between DRG constituting the L and the S “centers” of the uterine cervix afferent innervation (95.4 ± 2.0% and 42.3 ± 4.1%, respectively; Figure 6, *p* = 0.0003).

PNX-/CGRP-immunoreactive material was found in FB^+^ neurons belonging to all three size classes of sensory cells: small- and medium-sized cells were almost equally numerous (46.2 ± 5.7% vs. 47.5 ± 6.0%, respectively), while large-diameter perikarya were distinctly less numerous (6.3 ± 0.3%). Statistically significant differences were observed between the subsets of medium-sized DRG neurons located in the L and the S divisions of the spinal cord (52.2 ± 5.5% vs. 22.6 ± 7.6%, respectively; Table 6).

The majority of FB^+^/PNX^+^/CGRP^+^ neurons were unevenly scattered throughout the individual ganglionic subdomains. The largest population of such cells was found in the Cd domain (34.1 ± 2.5%), while slightly more than 20% of all FB^+^/PNX^+^/CGRP^+^ cells were located in the P and the Cr areas of the ganglia (22.3 ± 2.1%, 20.9 ± 1.6%, respectively); the less numerous populations of FB^+^/PNX^+^/CGRP^+^ perikarya were encountered in the Md and the Cn ganglionic domains (14.9 ± 3.2% and 7.8 ± 1.4%, respectively; Figure 7a). The distribution pattern of FB^+^/PNX^+^/CGRP^+^ neurons in the different domains of L and S DRG studied has been shown in Table 7.

#### 3.3.2. Characteristics of FB^+^ Neurons Co-Localizing PNX and SP

The second most numerous subpopulation of FB^+^/PNX^+^ DRG perikarya was formed by neurons containing simultaneously SP (Figure 5(B1–4)). Such coded cells accounted for 77.9 ± 1.0% of the total population of FB^+^/PNX^+^/SP^+^ UC-SNs found in all the DRG examined. Statistically significant differences in the numbers of FB^+^/PNX^+^/SP^+^ neurons were not observed between the left- and the right-sided DRG studied (75.5 ± 0.7% and 81.2 ± 2.4%, respectively); however, such differences were observed between the L and the S DRG forming the uterine cervix-supplying afferent “centres” (82.7 ± 2.1% and 63.7 ± 5.7%, respectively; Figure 6, *p* = 0.0358). In DRG from both the L and the S levels, the most numerous subgroup of FB^+^/PNX^+^/SP^+^ cells was formed by small-sized neurons (59.3 ± 3.3%), while such coded medium-sized perikarya were distinctly less numerous (36.6 ± 3.4%). In contrast, the retrogradely traced large-diameter neurons co-localizing PNX and SP were sporadically observed, constituting 4.1 ± 0.4%. Details concerning the relative frequency of FB^+^/PNX^+^/SP^+^ sensory neurons observed in the DRG from the L and the S levels of the spinal cord are shown in Table 8.

A clear marked accumulation of FB^+^/PNX^+^/SP^+^ neurons was found in the Cd ganglion domain (32.5 ± 2.3%), while such encoded UC-SNs were less numerous in the Cr and the P parts of the ganglia (24.9 ± 1.8%, 20.1 ± 1.5%, respectively). A smaller number of sensory neurons were found in the Cn and the Md part (14.2 ± 2.3%, 8.3 ± 2.0%, respectively) of DRG studied (Figure 7b). It is important to note that in the latter (Cr) domain, a statistically significant difference was observed in the number of FB^+^/PNX^+^/SP^+^ neurons between the L and the S DRG forming the uterine cervix-related afferent “centers” (17.8 ± 2.7% vs. 54.4 ± 6.7%, respectively). The details concerning the distribution pattern of FB^+^/PNX^+^/SP^+^ UC-SNs in the particular domains of DRG studied have been shown in Table 9.

#### 3.3.3. Characteristics of Neurons Co-Localizing PNX and nNOS

The third most abundant subpopulation of FB^+^/PNX^+^ cells consisted of neurons co-expressing nNOS (34.0 ± 0.9%; Figure 5(C1–4)). While such coded neurons were almost evenly distributed in both the left- and the right-sided DRG studied (35.0 ± 1.8% and 34.6 ± 1.8%), a clear, statistically significant difference was observed in the distribution of these cells between the DRG of the L and the S levels of the spinal cord (42.5 ± 2.6% vs. 13.3 ± 2.9%; Figure 6, *p* = 0.0018).

In DRG at both the L and the S levels, the nitrergic PNX^+^ cells supplying the uterine cervix were predominantly small- or medium-diameter cells (56.4 ± 2.6%, 35.7 ± 3.7%, respectively), whereas large-diameter cells constituted only a small proportion (7.6 ± 1.4%). Details concerning the relative frequency of FB^+^/PNX^+^/nNOS^+^ sensory neurons in the DRG examined from the L and the S levels of the spinal cord are shown in Table 10.

FB^+^/PNX^+^/nNOS^+^ sensory neurons were unevenly distributed inside the L and the S DRG, being mainly found in their Cd domain (29.8 ± 3.3%). The remainder of nNOS-containing FB^+^/PNX^+^ nerve cells were present in the P, Cr, Md, and the Cn (21.2 ± 4.1%, 20.1 ± 1.0%, 17.7 ± 3.6%, 11.2 ± 1.9%, respectively; Figure 7c) domains of DRG studied. However, it should be emphasized that the statistically significant differences in the number of these neurons were observed in the particular intraganglionic domains (for detail, see Table 11).

#### 3.3.4. Characteristics of Neurons Co-Localizing PNX and PACAP

In general, the colocalization of PACAP and PNX was found in 21.4 ± 1.3% (Figure 5(D1–D4)) of all retrogradely labeled, uterine cervix-supplying DRG neurons, being almost equally distributed in the left- as well as the right-sided ganglia (21.0 ± 2.3% and 22.4 ± 1.5%, respectively). Furthermore, differences observed in the numbers of FB^+^/PNX^+^/PACAP^+^ perikarya located in the L and the S DRG were statistically insignificant (22.8 ± 0.7% vs. 16.9 ± 3.7%, respectively; Figure 6).

When the diameter of sensory neurons was taken into consideration, the colocalization of PACAP and PNX was observed only in two size-classes of FB^+^ cells (total number of small- and medium-sized neurons, 61.3 ± 4.6% and 38.7 ± 4.6%, respectively), with large-diameter PNX^+^/PACAP^+^ perikarya being absent. Furthermore, as depicted in Table 12, distinct, statistically significant differences in their distribution pattern were observed between the L and the S DRG.

Focusing on the intraganglionic distribution pattern, the majority of FB^+^/PNX^+^/PACAP^+^ neurons in the L and S DRG were located in the P, Cr, and the Cd domains (29.2 ± 0.8%, 25.5 ± 1.1%, and 22.5 ± 1.2%, respectively), while the lowest numbers of these cells were observed in the Cn and the Md domains (12.4 ± 1.7% and 10.4 ± 2.9%, respectively; Figure 7d). It should, however, be emphasized that in the L DRG, a markedly lower number of FB^+^/PNX^+^/PACAP^+^ cells was found in the P subdomain, when compared with the same region of the S ganglia (26.6 ± 1.7% vs. 53.2 ± 7.0%; *p* = 0.0217). Detailed data concerning the distribution pattern of this subpopulation of the uterine cervix-supplying sensory neurons within different domains of investigated DRG have been summarized in Table 13.

#### 3.3.5. Characteristics of Neurons Co-Localizing PNX and GAL

The UC-SNs, colocalizing PNX and GAL, constituted a fairly small subfraction of all FB^+^/PNX^+^ DRG neurons (7.4 ± 1.6%; Figure 5(E1–4)), observed in the left and the right DRG (7.5 ± 0.5% and 6.6 ± 2.5%) and composed exclusively of small-diameter cells which were almost equally distributed in both the left and the right sided, as well as in both the L and the S DRGs studied (Figure 6, Table 14).

In general, FB^+^/PNX^+^/GAL^+^ sensory neurons were unevenly distributed between intraganglionic domains, being most numerous in the Cd and the Cr regions (34.7 ± 2.7% and 34.1 ± 3.7% of all traced PNX^+^/GAL^+^ neurons found, respectively; Figure 7e). The remainder of GAL-containing, FB^+^/PNX^+^ nerve cells were present in the Md, Cn, and the P (14.2 ± 5.4%, 11.1 ± 6.6% and 5.9 ± 3.0%, respectively) domains of DRG studied. However, while in the L DRG a few labeled neurons were found in all intraganglionic domains, the location of these cells in S DRG were restricted exclusively to the Cd and the Cr regions of the ganglia. Details concerning the distribution pattern of PNX^+^/GAL^+^ UC-SNs in the particular domains of L and the S DRG have been summarized in Table 15.

#### 3.3.6. Characteristics of Neurons Co-Localizing PNX and CRT

The co-incidence of PNX and CRT was found in an even smaller population of all DRG sensory neurons supplying the uterine cervix (3.1 ± 1.6%; Figure 5(F1–4)). These neurons were found exclusively in the L DRG, represented by small- (61.3 ± 5.6%) and medium-sized perikarya (38.7 ± 5.6%); (Table 16), and distributed relatively evenly in all subdomains of the ganglia, except for the Cn domain, where they were not observed (Figure 7f, Table 17). It should be pointed out once again that this subset of uterine cervix-supplying perikarya was absent in the S DRG (Figure 6).

#### 3.3.7. Characteristics of Neurons Co-Localizing PNX and SOM

The smallest subpopulation of FB^+^/PNX^+^ nerve cells found in this study (in the bilateral DRG, left-sided vs. right-sided, 0.3 ± 0.3% vs. 0.7 ± 0.3%, respectively) was the one whose neurons simultaneously contained SOM (0.7 ± 0.1% of all FB^+^/PNX^+^ neurons; Figure 5(G1–4)), and exclusively found in the small- and medium-sized perikarya (see Table 18, Figure 6).

Analysis of the intraganglionic distribution pattern of FB^+^/PNX^+^/SOM^+^ neurons has revealed that in the L DRG all of these cells were located in the Cn domain, while in the S DRG such coded neurons were evenly distributed in the Cd and the Cr domains (50 ± 21.8% and 50.0 ± 21.8%, respectively); (Table 19).

#### 3.3.8. The Pattern of Multiple Co-Localization of PNX with Biologically Active Substances in FB^+^ Neurons Innervating the Uterine Cervix

Double immunofluorescence technique combined with counting of FB^+^/PNX^+^ cell profiles on serial consecutive sections allowed the assessment of co-localization of more than two biologically active substances in the same cell, allowing us to draw following conclusions:(i)The vast majority of FB^+^/PNX^+^/CGRP^+^ neurons concurrently contained SP. These neurons also contained PACAP, GAL, CRT, and/or SOM. A significantly smaller number of retrogradely labeled PNX^+^/CGRP^+^ neurons also contained nNOS. In contrast, FB^+^/PNX^+^/CGRP^+^ nerve cells containing PACAP, CRT, or GAL constituted a significantly smaller subpopulation.(ii)CGRP was observed in the largest population of PNX^+^/SP^+^ sensory neurons supplying the uterine cervix. In contrast, neurons containing nNOS, PACAP, GAL, CRT, and SOM were found in much smaller numbers.(iii)The most numerous subpopulation of PNX^+^/NOS^+^ neurons supplying the uterine cervix was that which simultaneously showed immunoreactivity for PACAP. A slightly smaller subpopulation consisted of retrogradely labeled PNX^+^/nNOS^+^/CGRP^+^ neurons, while SP, GAL, and CRT were observed sporadically in such neurochemically coded nerve cells.(iv)In the case of FB^+^/PNX^+^/PACAP^+^ neurons, they most often additionally co-localized CGRP and/or SP. In turn, PNX^+^/PACAP^+^ neurons containing GAL, nNOS, or CRT were observed in much smaller numbers.(v)In retrogradely labeled PNX^+^ sensory neurons, GAL predominantly co-expressed with SP and CGRP. A significantly smaller subpopulation of FB^+^/PNX^+^/GAL^+^ neurons also contained PACAP or CRT, while nNOS was observed sporadically in such encoded neurons supplying the uterine cervix.(vi)The population of PNX^+^/CRT^+^ UC-SNs predominantly contained SP and CGRP. FB^+^/PNX^+^/CRT^+^ neurons containing PACAP or GAL were observed in much smaller numbers, while such encoded neurons additionally containing nNOS were observed sporadically.(vii)In the case of SOM, it was observed in a very small population of FB^+^/PNX^+^ neurons, mainly co-occurring with SP and CGRP.

Details concerning the relative percentages of individual subpopulations of retrogradely labeled PNX^+^ neurons are shown in Table 20.

## 4. Discussion

The uterus and uterine cervix are innervated, in addition to autonomic fibers, also by numerous sensory fibers [21,22]. Axonal tracing techniques combined with immunohistochemistry are used to examine the innervation and the distribution of neuropeptides in these afferent pathways in the rat [23,24], cat [25], as well as in the domestic pig [22,26].

Previous studies, based on injection of the FB tracer into the organ wall, have revealed that the uterine cervix-supplying autonomic and sensory nerve terminals originate from L and S paravertebral ganglia, inferior mesenteric ganglia, paracervical ganglia as well as from DRG of Th10-L4 and S2-S3 levels of the spinal cord [22], which was corroborated by the results of the present study (see above). When the relative frequencies of retrogradely labeled neurons in the particular DRG were taken into consideration, a significant number of traced cells were located in some of the examined L and S DRG, forming specific “sensory supply centers” of the uterine cervix; these were, respectively, DRG L3 and L4 in the L spinal cord, and DRG S3 and S4 in the S spinal cord. On the other hand, only single FB^+^ perikarya were found in DRG of the thoracic spinal cord and, in contrast to the results of Wa̧sowicz and colleagues (1998), also in the L5 DRG [22].

Furthermore, when the tracer was injected into the paracervical region (right side of the uterus), the presence of FB^+^ neurons was observed in the ipsilateral DRG of exactly the same levels, strongly corroborating the suggestion that all sensory fibers supply the uterine cervix transit through the paracervical ganglia [22].

When comparing the localization of this population of retrogradely labeled sensory neurons in the spinal ganglia of the rat (L1, L2, L6, S1; [2]), cat (L3, L4, S1–S3; [3]), and the domestic pig (L3, L4, S3, S4; [22], this study), distinct interspecies differences in the involvement of DRG at individual spinal cord levels in the sensory innervation of the uterine cervix can be observed. While in the pig the majority of the uterine cervix-supplying FB^+^ neurons were located in the S DRG, with a lesser representation in the L DRG (approximately 64% and 35% of all traced cells, respectively), such cells, receiving sensory stimuli from the rat cervix are located more rostral, and, in the cat, they form a distinct “sensory center” consisting of S1-S3 DRG.

Moreover, in the present study we also attempted to determine whether the uterine cervix-supplying sensory PNX^+^ neurons show any tendency towards somatotopic distribution in the individual domains of the ganglia examined (see Figure 1), as previously described for the innervation pattern of the rat hind paw digits, the skin, and other tissues of the cat [27], or the intraganglionic organization of the human [28] or feline DRG [29]. Analysis of the spatial distribution of labeled PNX^+^ cells in the subdomains of the studied DRG indicated that the majority of these neurons were located in the Cd domain, which may suggest a rostro-caudal organization of the reproductive organs’ innervation. However, further studies using different combinations of retrograde tracers (e.g., FB in combination with FluoroGold, Nuclear Yellow, etc.) are necessary to confirm or refute this hypothesis.

PNX, since its discovery in 2013 [1], has attracted increasing interest due to its involvement in the regulation of sensory and reproductive system functions. As our preliminary results demonstrated the existence of relatively numerous PNX^+^ nerve terminals (a significant proportion of which also contained SP, a “marker substance” of afferent fibers) in the uterine cervix wall, especially in the submucosal and muscular layers, this prompted us to conduct detailed tracing and immunochemical studies focused on the neurochemical architecture of the DRG neurons involved in the uterine cervix innervation, with special regard to the pattern(s) of PNX co-occurrence with other neurotransmitters found previously in the porcine DRG cells. In general, approximately 33% of all uterine cervix-supplying afferent neurons located in the lumbosacral DRG were PNX^+^ which suggests the significant physiological importance of this neuropeptide in the regulation of cervical function. Furthermore, the co-occurrence of various combinations of other biologically active substances in PNX^+^ neurons further indicates the involvement of PNX in the functioning of various regulatory pathways in the organ.

### 4.1. Co-Localization and Putative Physiological Relevance of PNX and CGRP

The most abundantly represented in the entire population of retrogradely labeled PNX^+^ cells, found in the present study, were neurons co-containing CGRP (approximately 81%), constituting a significant portion of all PNX^+^ cells in porcine lumbosacral DRG, as shown by previous studies (CGRP co-existence was observed in over 96% of all PNX^+^ cells in these ganglia; [16]). CGRP is a neuropeptide released from sensory nerve terminals under inflammatory conditions [30] and plays a key role in nociceptive transmission within the spinal cord [31]. Acting synergistically with SP, CGRP is a potent vasodilator [32,33], contributing to the regulation of blood flow in the female reproductive tract and has been implicated in the process of cervical ripening [34]. In pregnant rats, increased levels of CGRP and its mRNA are observed in the lumbosacral (L6–S1) DRG and spinal cord, while CGRP expression in the cervix itself decreases. Moreover, it has been shown that CGRP mRNA expression in DRG neurons responds to estrogen stimulation in a dose-dependent manner, and this response can be attenuated using estrogen receptor antagonists. These findings collectively suggest that estrogen modulates CGRP expression in lumbosacral sensory neurons, thereby potentially influencing both central and peripheral mechanisms involved in cervical remodeling during parturition [35].

It is also worth mentioning that the percentage of FB^+^/PNX^+^ cells co-expressing CGRP differed significantly, depending on the spinal cord level—while at the L level such colocalization was observed in approximately 95% of PNX^+^ perikarya, at the S level only 42% of PNX^+^ cells co-contain simultaneously CGRP. Such a significant difference may suggest a functional heterogeneity of the population of PNX^+^/CGRP^+^ neurons innervating the cervix. The higher percentage of PNX^+^/CGRP^+^ cells at the L level may suggest that this segment of the spinal cord plays a more significant role in processing sensory stimuli related to nociception, blood flow modulation, or neurogenic inflammation in the cervix. In line with our previous findings [16], the majority of PNX^+^/CGRP^+^ cells in the porcine DRG were small-sized neurons, which additionally corroborates the suggestion of their nociceptive nature.

### 4.2. Co-Localization and Putative Physiological Relevance of PNX and SP

The second-most abundant PNX^+^ subpopulation of afferent DRG neurons involved in the sensory innervation of the uterine cervix was the subset of SP co-expressing cells. This is consistent with previous reports regarding the involvement of SP^+^ neurons in the innervation of peripheral tissues, including the cervix [36,37]. It has also been suggested that this tachykinin, upon local release, is able to cause neurogenic pain and inflammation [38]. Beyond its well-established role in nociceptive signaling and inflammatory responses, SP has also been implicated in the regulation of pubertal development and female fertility [39]. It has also been shown that the highest concentration of SP is found in the vagina, followed by the uterine cervix, endometrium, and ovaries, and that its levels were significantly reduced during pregnancy in all reproductive organs but, surprisingly, significantly increased after ovariectomy [40]. Interestingly, administration of SP to ovariectomized rats mimics aspects of the neurogenic inflammation observed in the parturient cervix, causing recruitment of immune cells, vasodilation, and plasma extravasation, suggesting that SP may play an important role in cervical maturation [36] and may be involved in tissue rearrangement [38].

In contrast to the uterine cervix of the rat, where SP level declines in the last trimester of pregnancy, an increase in both SP and its mRNA levels is observed in the L6-S1 DRG neurons, with the expression of the latter showing an estrogen dose dependency [35,41,42]. Thus, increase in the estrogen receptor density in the DRG neurons during parturition may lead to an increase in SP synthesis and this peptide, released from cervical sensory nerve endings, binds to NK1 receptors to participate in cervical remodeling [35].

It is also interesting and worth mentioning that the co-occurrence and, consequently, cooperation of PNX and SP in the regulation of reproductive organ functions, including the uterine cervix, seems not to be limited only to the level of peripheral interactions, but both substances, co-occurring in central neurons, may modulate the function of the hypothalamic–pituitary–gonadal (HPG) axis. Known for its involvement in the transmission of pain and proinflammatory stimuli, SP may also influence hypothalamic GnRH secretion, in this way indirectly modulating pituitary and gonadal activities: while on the one hand SP deficiency led to delayed sexual maturation and/or to infertility in females [39], on the other hand the activation of NK1 receptor by SP during the prepubertal period accelerated the onset of sexual maturation, possibly through early activation of kisspeptin-containing neurons [39]. In turn, PNX, a neuropeptide with documented effects on reproduction, regulates GnRH expression and may modulate HPG axis function, affecting follicle maturation, ovulation, and hormonal preparation of the cervix for childbirth. The synergistic action of SP and PNX may therefore “integrate” peripheral signals (e.g., from the cervix) with central neuroendocrine regulation, which is of particular importance during pregnancy, childbirth, as well as the estrous cycle.

### 4.3. Co-Localization and Putative Physiological Relevance of PNX and nNOS

Three NOS isoforms are expressed in the rat cervix: eNOS, iNOS, and nNOS. In addition, while particularly pronounced expression of eNOS and iNOS has been observed in the uterine body [43], primarily nNOS expression was observed in the ovaries [44]. The presence of nNOS has not only been found in nitrergic nerve terminals of the uterine cervix of both pregnant and non-pregnant women, but also in the intermediate and superficial layers of epithelial cells. Cervical glands and vascular endothelial cells remained free of expression of the enzyme [45]. Increased expression of the nNOS isoform in the uterine cervix structures has been observed during childbirth in humans [46]. Similar phenomena were observed in rodents, where NO induced cervical maturation [47], and its release from this area was also markedly elevated during labor [43]. It seems likely that PNX may participate in the regulatory effects described above by modulating nNOS activity in a currently unknown way.

Changes in the number of nitrergic cells in the DRG neuron population were observed both during development (e.g., in ovine fetuses, the percentage of nitrergic neurons, observed in DRG was 94% on day 60 and dropped to about 30% on day 90 of the fetal life, with no apparent further changes until the postnatal period [48], as well as in response to pathological stimuli (e.g., cutting or crushing of the nerve with a ligature [1,49], respectively); these observations suggest that the presence of nNOS in afferent cells may reflect the role of nitric oxide in the regulation of developmental [48] as well as repair processes, in which this gaseous transmitter plays a neuroprotective role [1,50]. It should be emphasized, however, that nitric oxide, acting as a signaling molecule at both central and peripheral sensory terminals, may play an important role in transmission of visceral sensation [51]. PNX has also been implicated in conveying visceral pain sensations [1]. Thus, since colocalization of PNX and nNOS was observed in the present study in approximately 34% of all retrogradely labeled DRG perikarya, both molecules may collaborate in the trafficking of sensory modalities from pelvic viscera, especially the cervix or uterine body. Also noteworthy is the high degree of colocalization of PNX, nNOS, CGRP, and SP in retrogradely labeled DRG cells supplying the uterine cervix, which implies much more complex mechanisms of nociceptive transmission from the organ than previously thought. However, further research is necessary to thoroughly understand the importance and interactions between these substances in regulating cervical function.

### 4.4. Co-Localization and Putative Physiological Relevance of PNX and PACAP

The next most abundant population of PNX^+^ sensory neurons, as shown by double immunofluorescence studies, were neurons containing PACAP (21.4%). PACAP is found mainly in small DRG neurons, where it coexists with other neuropeptides such as CGRP, [52] and in the superficial layers of the posterior horn of the rat spinal cord [53]. In the pig, PACAP was present mainly in small-sized neurons [16], though in the subpopulation subserving the sensory pathway(s) of the uterine cervix—mainly in small- and medium-sized cells. In the rat, PACAP was also found mainly in small neurons, but after axotomy its expression in DRG neurons of each diameter increased very quickly, which may suggest its participation in the early phase of the adaptive response [54]. Although its exact roles in DRG neuronal survival and differentiation are not fully understood as of now, as may be judged from the available literature, PACAP protects neurons from death, stimulates the growth of neural processes, and induces CGRP expression—in a manner that is partly similar to the action of nerve growth factor (NGF) [55]. During pregnancy, PACAP expression shows developmental stage-dependent variability—in rats, an initial increase in PACAP levels is observed, but its expression decreases in late pregnancy [56]. These changes were also reflected on the levels of DRG neurons and the dorsal horns of the spinal cord (PACAP synthesis increased in DRG (L6-S1) neurons while simultaneously decreased in the spinal cord in late pregnancy). Interestingly, PACAP mRNA levels in L6-S1 DRG neurons increased with estrogen treatment [56]. Importantly, under inflammatory conditions, PACAP and its receptor (PAC1) mRNA levels significantly decreased in the uterus [57], while in DRG cells they tend to increase [26], which may indicate its involvement in the neuroimmune response to inflammation in the female reproductive organs. Considering the evidence indicating the neuroprotective and modulatory properties of PNX—including its involvement in visceral pain processing and inflammatory response—it is possible that both peptides participate in the synergistic modulation of neuronal plasticity and nerve regeneration under conditions of stress, injury, or inflammation.

### 4.5. Co-Localization and Putative Physiological Relevance of PNX and GAL

GAL and its receptors are widely expressed in the central and peripheral nervous system and are involved in an in-deep modulation of neurotransmission in numerous neural pathways [16,58]. GAL^+^ terminals were also found in numerous peripheral tissues, including reproductive organs [59]. Regarding the galaninergic innervation of the uterine cervix in the rat, few GAL^+^ terminals were found in the cervical wall, which, as shown by retrograde tracing studies, are peripheral processes of a small group of small-diameter sensory perikarya observed in DRG of L6 and S1 levels [60]. This is well in line with the present study, in which afferent neurons immunoreactive to PNX and GAL were found in a small fraction (7.4%) of exclusively small-sized cells of the lumbosacral DRG. As GAL^+^ fibers supplying the cervical wall were observed almost exclusively in the myometrium and in the periarterial location, it seems likely that the PNX and GAL tandem controls the function of vascular and nonvascular smooth muscle, as previously suggested for GAL [16,59,61]. However, to fully understand the role of PNX and/or GAL, further studies are necessary, particularly those focusing on the tissue distribution of the PNX receptor (GRP173) and the pharmacological effects of PNX on uterine cervical smooth muscle behavior [59].

Moreover, it should also be noted that GAL expression significantly increases in DRG cells affected by pathological conditions. As shown in the rat, uterine inflammation results in an increase in the density of GAL^+^ fibers in the myometrium (which is explained by the participation of this substance in the increased activity of the muscle layer, which is supposed to lead to faster removal of inflammatory exudate from the uterine cavity; [62]). In turn, damage to the integrity of GAL^+^ neuronal processes, for example, due to extirpation of the uterus (as their target organ), leads to a dramatic increase in the number of sensory cells supplying the organ, which de novo start to synthesize GAL, probably as a neuroprotective substance, allowing the neurons to survive the injury until they find replacement target tissues [42,62]. However, further studies on the plasticity of PNX expression under the above-mentioned conditions are necessary to fully understand its (possible) importance in mitigating the effects of inflammation or physical tissue damage and, consequently, the integrity of sensory neuronal projections [16,61].

### 4.6. Co-Localization and Putative Physiological Relevance of PNX and CRT

Previous studies have shown that PNX and CRT coexist in a relatively small subpopulation of DRG cells (approximately 10% of all PNX^+^ neurons), the vast majority of which were small in diameter [16]. In the present study, the sensory neurons coded in this way, retrogradely labeled from the uterine cervix, constituted a marginal subpopulation of all FB^+^ sensory perikarya examined (approximately 3%). This is somewhat in line with observations made in the rat, in which species CRT^+^ DRG neurons also constituted about 10% of all spinal ganglia cells. However, it should be emphasized that there are clear interspecies differences both in the distribution of CRT^+^ neurons in the DRG of individual spinal levels and in the presence of this calcium ion homeostasis regulating protein in neurons belonging to different size classes. However, it should be emphasized that there are clear interspecies differences both in the distribution of CRT^+^ neurons in the DRG of individual spinal levels and in the presence of this calcium ion homeostasis-regulating protein in neurons belonging to different size classes; while in the rat CRT was found in neurons of both L and S DRG [63], in the pig CRT^+^ cells were present exclusively in the DRG of L levels. Moreover, in the rat, CRT^+^ neurons belonged mostly to the class of cells with medium or large diameter, whereas in the pig, CRT was not observed at all in large-diameter perikarya [63]. As it may be judged by the unique pattern of CRT distribution in the spinal cord (including dense fiber networks in Clarke’s column and strong staining in superficial laminae, especially lamina II), CRT appears to be involved in the functioning of the nociceptive and proprioceptive pathways [64]. In the rat, CRT-immunoreactive nerves were present in the uterine horns and cervix where they were associated with arteries, uterine smooth muscle, glands, and the epithelium [63]. CRT is also defined as a marker of properly functioning endometrial stroma, and its altered expression may indicate disturbances in endometrial physiology, especially in cases of abnormal uterine bleeding [65].

CRT, synthesized in neurons, may act as a binder of intracellular free calcium, helping to regulate its concentration [66]. It appears that the early high expression of CRT in DRG cells suggests its transient role in the regulation of intracellular calcium ion concentration early in embryonic development, before neurotrophins take over the precise control of calcium homeostasis. This biphasic pattern of CRT expression (between day 11 and 14 of fetal life approximately 75% of all DRG neurons expressed CRT, while around day 17 of fetal life only 8% sensory neurons was still CRT^+^) may reflect the changing functional needs of developing neurons during the maturation of the nervous system [67,68].

### 4.7. Co-Localization and Putative Physiological of PNX and SOM

The last, smallest population of retrogradely labeled PNX^+^ cells observed in this study were neurons additionally expressing SOM, an inhibitory neuropeptide expressed in a small population of DRG neurons in both rat and pig (up to 10% of the whole population of sensory DRG cells [16,69]). Our studies showed that a very small fraction of FB^+^/PNX^+^/SOM^+^ DRG neurons (approximately 0.7%) innervate the uterine cervix in the pig. Moreover, analyzing the available literature, it is currently almost impossible to assign this fraction of DRG cells and their possible physiological role(s) in the cervix. On the one hand, both SOM and PNX seem to be deeply involved in the transmission and regulation of itch sensation [70,71]: the ablation of primary afferent neurons expressing SOM led to deficits in itch sensation [71], while the intrathecal administration of SOM induced scratching behavior [72]. On the other hand, SOM seems to play an important role in thermal nociception, as mice with conditional deletion of SOM expression in DRG cells showed significantly increased sensitivity to thermal noxious stimuli [73]. However, at present there is a lack of not only data but also any suggestions as to the possible physiological role(s) of SOM and PNX in the sensory supply of the uterine cervix. This clear gap in our knowledge of the functions and physiological significance of individual DRG cell subpopulations requires attention and in-depth research to provide a satisfactory explanation.

### 4.8. Co-Occurrence Pattern of Multiple Biologically Active Substances in PNX-Containing UC-SNs

In this study, we also attempted to answer the question whether PNX-positive DRG sensory neurons involved in the cervical innervation can simultaneously contain more than two of the analyzed substances. Analysis of the co-occurrence patterns of the studied neurotransmitters, performed on consecutive serial cryostat sections, provided important and new information, expanding our knowledge of the chemical phenotype of sensory neurons innervating the uterine cervix. At the same time, however, the results obtained revealed numerous issues requiring further explanation, which will require in-depth studies of both qualitative and functional nature. Of particular interest is the chemical profile of a large subpopulation of PNX^+^ neurons, which conduct sensory information from the uterine cervix. These neurons show diverse combinations of nNOS, PACAP, GAL, CRT, and/or SOM, co-occurring with key sensory transmitters, such as SP and/or CGRP. Although the issue of joint or alternating release of two neurotransmitters by sensory neurons has been described in numerous studies, there is a lack of data on the physiological significance and potential mechanisms of simultaneous release of more transmitters from a single nerve ending to the target tissue. Interpretation of the obtained results is also hampered by the lack of knowledge about specific neuronal subtypes and about the distribution and characteristics of receptors for individual transmitters in effector tissues. Therefore, without further detailed functional studies, it is impossible to clearly link the specific sensory modalities to the particular DRG subpopulations differing in their neurotransmitter profile.

### 4.9. Unanswered Questions and Possible Further Research Directions Resulting from Present Experiment

(1)Although the coexistence of PNX with substances generally considered to be pain transmitters (e.g., SP, CGRP, GAL, and/or PACAP) strongly supports the authors’ hypothesis of the co-participation/modulation of nociceptive transmission by the PNX^+^ subpopulation of UC-SNs, it is necessary to conduct studies focusing on the “receptor landscape” of both UC-SNs and their target tissues in the innervated organ. Particular attention should be paid to the TRPV family receptors (especially the TRPV1 and TRPA1 ion channels), as they are not only localized in the uterine cervix of humans and other animal species, but have also been shown to be variably expressed and activated during pregnancy, parturition, and/or pathological conditions of the uterus [74,75]. The need to further explore this issue is supported by data indicating a significant influence of sex hormones on their expression pattern [76]; therefore, examining the co-occurrence of PNX, its receptor (GRP173) and members of the TRP family will allow us to specify and validate the potential role of PNX in the transmission of pain stimuli from the uterine cervix.(2)Another extremely interesting and potentially very important issue from the clinical point of view is to explain whether, and if so, how the transmission substances released from the sensory endings are able to influence both the behavior (growth and progression) and their microenvironment of uterine cervix tumors. As may be judged from the available literature, interactions between the sensory nervous system and tumors should become the focus of further research, as observations from a growing number of research groups suggest a new and potentially very interesting pathway for pharmacologically influencing the biology of tumors in various organs (for the most recent reviews, see [77,78,79]. Thus, understanding these relationships and identifying potential pharmacological targets could significantly expand our therapeutic options in the future.(3)Last, but not least, there is also a great need to conduct in-depth pharmacological studies on the possible interactions of biologically active substances co-occurring in UC-SNs and their combined effects on the target tissues of the studied neurons.

## 5. Conclusions

To date, there has been little information in the available literature on the neurochemical coding of sensory neurons that use PNX as one of their transmitters [16]. This study, utilizing analysis of the presence of a range of neurotransmitter molecules in serial profiles of PNX^+^ DRG cells involved in the sensory supply of the uterine cervix, expands our understanding of the innervation of the genital tract, demonstrating that the chemical coding of the neurons contributing to these neural pathways is much more complex than previously thought. These results imply the need for further research, particularly pharmacological and functional studies, which should allow for a better understanding of the physiological roles and the significance of co-occurring and interacting biologically active substances in the regulating of the uterine cervix function.

## Figures and Tables

**Figure 1 cells-14-01847-f001:**
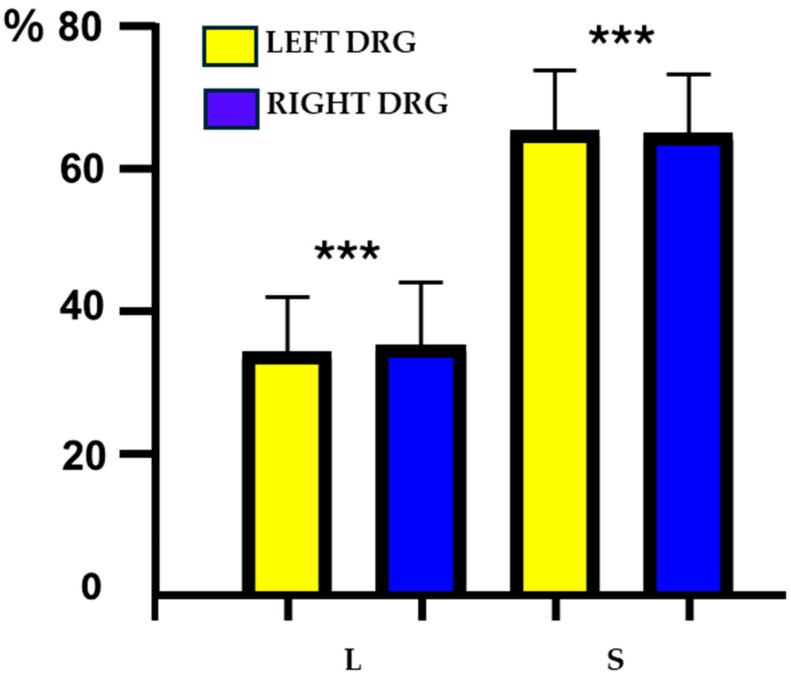
Bar diagram showing the relative percentages of FB-positive (FB^+^) sensory neurons located in the dorsal root ganglia (DRG) of the left and the right lumbar (L) and sacral (S) segments of the spinal cord. The data obtained were pooled in all the animals and presented as mean ± standard error of measurement (SEM), referring to the number of animals (*N* = 6). Asterisks mark statistically significant differences: *** *p* = 0.0002.

**Figure 2 cells-14-01847-f002:**
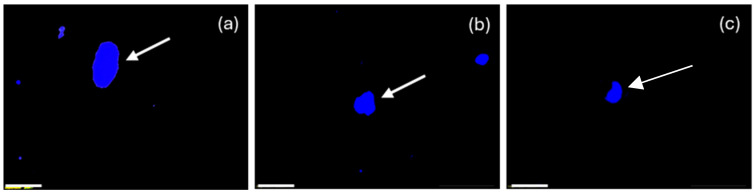
Representative images of the FB^+^ nerve cells in the porcine DRG belonging to the different size classes; (**a**) large cell, (**b**) medium-sized cell, (**c**) small-sized cell. The blue profiles visible in (**a**,**b**) which are not indicated by an arrow, are fragments of other traced UC-SNs, cut in a different plane than the one corresponding to their largest diameter. Bar in all the images—50 μm.

**Figure 3 cells-14-01847-f003:**
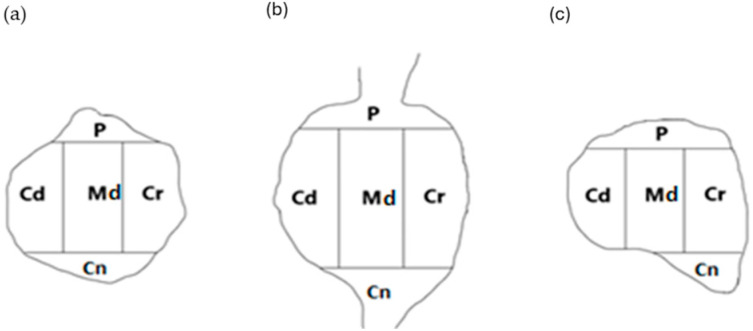
A schematic diagram of DRG section showing its arbitrary division into topographical domains, in which the occurrence and relative frequency of FB^+^ sensory neurons were studied: Cn—central domain, Cd—caudal domain, Cr—cranial domain, P—peripheral domain, Md—middle ganglion area; section from the (**a**) proximal, (**b**) middle, and (**c**) distal part of the ganglion.

**Figure 4 cells-14-01847-f004:**
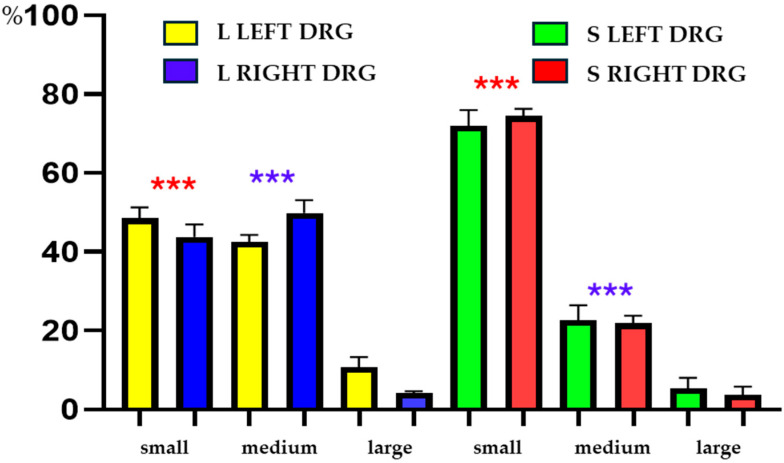
The bar graph showing relative frequencies (%) of subpopulations sizes of PNX^+^ and retrogradely labeled neurons located in the individual left and the right DRG of the L and S levels of spinal cord. The data obtained were pooled for each animal and are presented as mean ± SEM. Asterisks mark statistically significant differences: ******* *p* = 0.004, ******* *p* = 0.0009.

**Figure 5 cells-14-01847-f005:**
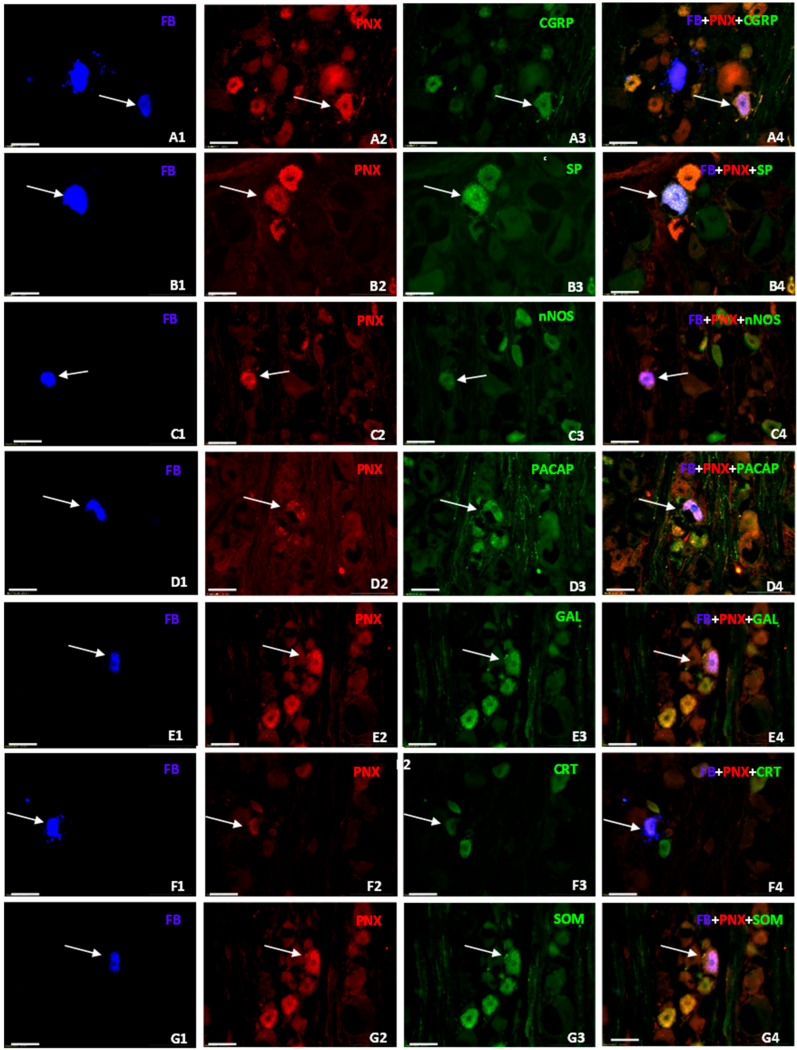
Representative images of DRG neurons supplying the uterine cervix wall in the domestic pig. The images were taken separately from blue (**A1**, **B1**, **C1**, **D1**, **E1**, **F1**, **G1**), red (**A2**, **B2**, **C2**, **D2**, **E2**, **F2**, **G2**), and green (**A3**, **B3**, **C3**, **D3**, **E3**, **F3**, **G3**) fluorescent channels. Pictures **A4**, **B4**, **C4**, **D4**, **E4**, **F4**, and **G4** represent images where blue, red, and green channels were digitally superimposed. Arrows represent FB+ neurons (**A1**, **B1**, **C1**, **D1**, **E1**, **F1**, **G1**) that were simultaneously PNX (**A1**, **A2**, **B2**, **C2**, **D2**, **E2**, **F2**, **G2**), CGRP (**A3**), SP (**B3**), nNOS (**C3**), PACAP (**D3**), GAL (**E3**), CRT (**F3**), or SOM (**G3**). Bar in all the images—50 μm.

**Figure 6 cells-14-01847-f006:**
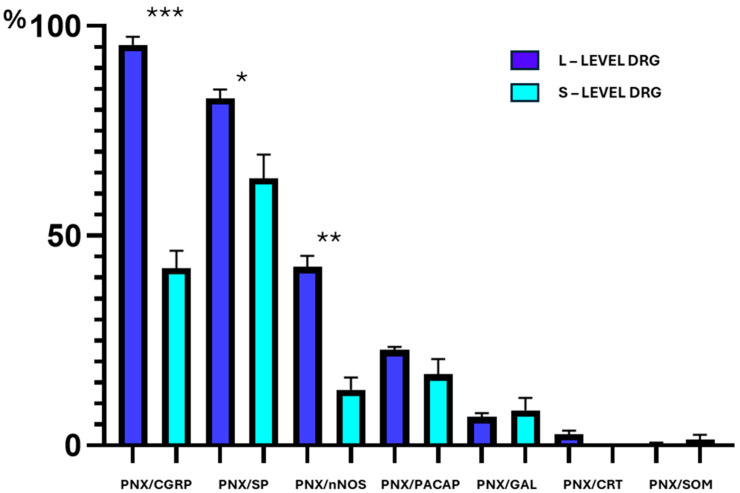
The relative frequency of PNX^+^ UC-SNs colocalizing the particular neurotransmitters studied in the L (dark blue bars) and the S (light blue bar) segments of DRG. The data obtained were pooled in all the animals and presented as mean ± SEM, referring to the number of animals *(N* = 6). Asterisks mark statistically significant differences: * *p* = 0.0358; ** *p* = 0.0018; *** *p* = 0.0003.

**Figure 7 cells-14-01847-f007:**
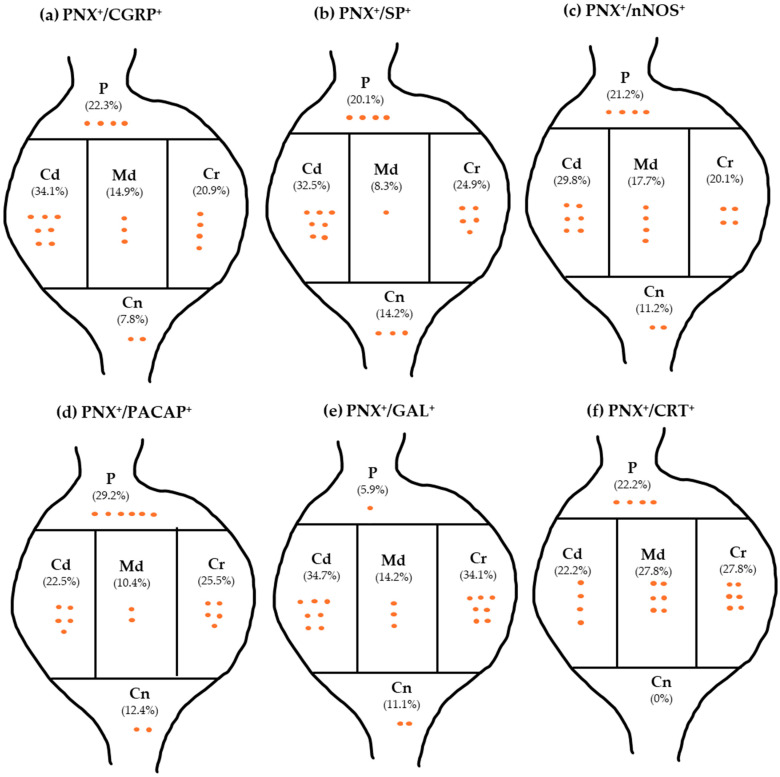
The relative frequency of PNX^+^ UC-SNs colocalizing particular neurotransmitters in the different ganglionic domains of DRG studied. Each orange dot in the figure represents approximately 5% of the cell population. The data obtained were pooled in all the animals and presented as mean ± SEM, referring to the number of animals (*N* = 6).

**Table 1 cells-14-01847-t001:** The relative frequency of FB^+^ neuronal profiles located in the left and right DRG of Th, L and S levels of the spinal cord. The data obtained were pooled and presented as mean ± SEM, referring to the number of animals (*N* = 6).

Segments of the Spinal Cord %												
DRG	Th12	Th13	Th14	Th15	L1	L2	L3	L4	L5	S2	S3	S4
Left	0.2 ± 0.2	0.2 ± 0.1	0.2 ± 0.1	0.4 ± 0.3	1.1 ± 0.6	4.7 ± 0.1	10.3 ± 2.6	14.5 ± 3.4	2.2 ± 0.4	0.4 ± 0.4	37.6 ± 4.4	28.2 ± 7.8
Right	0.0 ± 0.0	0.0 ± 0.0	0.0 ± 0.0	0.4 ± 0.4	1.6 ± 1.2	4.5 ± 2.0	12.3 ± 0.8	16.0 ± 1.4	2.8 ± 0.9	0.7 ± 0.6	32.8 ± 5.4	28.9 ± 3.5

**Table 2 cells-14-01847-t002:** Percentages of FB^+^ neuronal populations of different size classes (small, medium, large) located in the left and the right DRG of the L and S segments of the spinal cord. The data obtained were pooled in all the animals and presented as mean ± SEM, referring to the number of animals (*N* = 6). Asterisks mark statistically significant differences: ****** *p* = 0.0041; ***** *p* = 0.0274.

Level of the Spinal Cord %						
L				S		
DRG	Small *	Medium	Large **	Small *	Medium	Large **
Left	37.1 ± 7.6	41.3 ± 4.4	21.6 ± 3.2	59.0 ± 3.8	35.2 ± 3.0	5.8 ± 1.3
Right	34.8 ± 6.9	47.6 ± 3.4	17.6 ± 3.6	55.2 ± 3.5	40.0 ± 3.0	4.8 ± 0.5

**Table 3 cells-14-01847-t003:** Percentages of the intraganglionic distribution pattern of the population of FB^+^ neurons located in the left and the right DRG studied taken from the L and the S levels of the spinal cord. Data obtained from all the animals were combined and presented as mean ± SEM, referring to the number of animals (*N* = 6). Asterisks mark statistically significant differences: * *p* = 0.0471; ****** *p* = 0.0028; ******
*p* = 0.0033.

Level of the Spinal Cord %
L	S
DRG	Cn *	Cd **	Cr **	P	Md	Cn *	Cd **	Cr **	P	Md
Left	13.4 ± 0.4	36.5 ± 1.2	22.4 ± 0.5	15.1 ± 0.3	12.6 ± 1.6	6.6 ± 1.0	22.2 ± 2.8	38.0 ± 4.3	20.0 ± 1.9	13.2 ± 1.2
Right	8.9 ± 1.9	32.9 ± 1.2	22.8 ± 1.8	23.9 ± 1.7	11.5 ± 2.6	5.4 ± 0.4	21.0 ± 2.6	46.2 ± 2.4	17.1 ± 3.0	10.3 ± 1.1

**Table 4 cells-14-01847-t004:** Percentages of the population of FB^+^/PNX^+^ neurons of different sizes (small, medium, large) located in the left and the right DRG of L and S segments of the spinal cord. The data obtained were combined for all animals and presented as mean ± SEM, referring to the number of animals (*N* = 6). Asterisks mark statistically significant differences: ******* *p* = 0.004, ******* *p* = 0.0009.

Segments of the Spinal Cord (%)						
L				S		
DRG	Small ***	Medium ***	Large	Small ***	Medium ***	Large
Left	47.4 ± 1.6	42.5 ± 2.9	10.1 ± 2.1	71.9 ± 4.0	22.8 ± 3.7	5.3 ± 2.6
Right	42.1 ± 0.7	50.9 ± 3.2	7.0 ± 1.7	74.5 ± 1.8	21.9 ± 1.8	3.6 ± 2.1

**Table 5 cells-14-01847-t005:** The intraganglionic distribution pattern of FB^+^/PNX^+^ neuronal population, located in the particular subdomains of DRG from the L and S segments of the spinal cord. The data obtained were pooled for each animal (*N* = 6) and are presented as mean ± SEM. Asterisks mark statistically significant differences: ** *p* = 0.0030, ** *p* = 0.0013.

Segments of the Spinal Cord (%)										
L						S				
	Cn **	Cd	Cr **	P	Md	Cn **	Cd	Cr **	P	Md
Left	14.5 ± 0.8	33.3 ± 2.0	19.5 ± 3.7	20.7 ± 2.8	12.0 ± 1.5	7.8 ± 1.3	33.3 ± 6.1	29.3 ± 4.9	19.2 ± 1.3	10.4 ± 2.9
Right	14.2 ± 1.7	32.3 ± 3.3	16.0 ± 1.5	20.5 ± 3.8	17.0 ± 2.0	4.4 ± 1.1	35.5 ± 5.7	33.5 ± 4.2	18.0 ± 2.9	8.6 ± 1.0

**Table 6 cells-14-01847-t006:** Tabulated summary of the relative frequency (%) of FB^+^/PNX^+^/CGRP^+^ cells in the L and the S DRG examined depending on their diameter. Data obtained from all the animals studied were pooled and presented as mean ± SEM, referring to the number of animals (*N* = 6). Asterisks mark statistically significant differences: *** ***p* = 0.0376.

Segment of the Spinal Cord (%)					
L			S		
Small	Medium *	Large	Small	Medium *	Large
41.6 ± 5.7	52.2 ± 5.5	6.2 ± 0.7	72.2 ± 8.6	22.6 ± 7.6	5.2 ± 2.8

**Table 7 cells-14-01847-t007:** The relative frequency (%) of FB^+^/PNX^+^/CGRP^+^ neurons in the particular intraganglionic domains of L and the S DRG studied. Data obtained from all the animals studied were pooled and presented as mean ± SEM, referring to the number of animals (*N* = 6).

Segment of the Spinal Cord (%)									
L					S				
Cn	Cd	Cr	P	Md	Cn	Cd	Cr	P	Md
8.7 ± 1.5	33.9 ± 2.9	18.5 ± 0.9	23.0 ± 2.9	15.9 ± 3.3	2.4 ± 2.2	37.7 ± 2.2	37.8 ± 8.0	15.5 ± 4.4	6.6 ± 3.8

**Table 8 cells-14-01847-t008:** Tabulated summary of the relative frequency (%) of FB^+^/PNX^+^/SP^+^ cells in the L and the S DRG examined depending on their diameter. Data obtained from all the animals studied were pooled and presented as mean ± SEM, referring to the number of animals (*N* = 6).

Segment of the Spinal Cord (%)					
L			S		
Small	Medium	Large	Small	Medium	Large
55.1 ± 5.7	40.5 ± 5.2	4.4 ± 1.7	69.3 ± 8.6	27.7 ± 1.3	3.0 ± 3.0

**Table 9 cells-14-01847-t009:** The relative frequency of FB^+^/PNX^+^/SP^+^ neurons in the particular intraganglionic domains of the L and the S DRG studied. Data obtained from all the animals studied were pooled and presented as mean ± SEM, referring to the number of animals (*N* = 6). Asterisks mark statistically significant differences: ****** *p* = 0.0071.

Segment of the Spinal Cord (%)									
L					S				
Cn	Cd	Cr **	P	Md	Cn	Cd	Cr **	P	Md
9.6 ± 1.9	34.7 ± 3.4	17.8 ± 2.7	21.2 ± 1.6	16.7 ± 2.7	2.9 ± 2.9	24.1 ± 2.0	54.4 ± 6.7	13.3 ± 1.3	5.3 ± 3.9

**Table 10 cells-14-01847-t010:** Tabulated summary of the relative frequency (%) of FB^+^/PNX^+^/nNOS^+^ cells in the L and the S DRG examined depending on their diameter. Data obtained from all the animals studied were pooled and presented as mean ± SEM, referring to the number of animals (*N* = 6).

Segments of the Spinal Cord (%)					
L			S		
Small	Medium	Large	Small	Medium	Large
54.3 ± 2.9	37.6 ± 5.0	8.1 ± 2.5	66.7 ± 9.6	27.7 ± 5.5	5.6 ± 5.5

**Table 11 cells-14-01847-t011:** The relative frequency of FB^+^/PNX^+^/nNOS^+^ neurons in the particular intraganglionic domains of the L and the S DRG studied. Data obtained from all the animals were pooled and presented as mean ± SEM, referring to the number of animals (*N* = 6). Asterisks mark statistically significant differences: ****** *p* = 0.0052 ***** *p* = 0.0151; ***** *p* = 0.0337; ****** *p* = 0.0094.

Segments of the Spinal Cord (%)									
L					S				
Cn **	Cd *	Cr *	P	Md **	Cn **	Cd *	Cr *	P	Md **
12.2 ± 2.1	28.2 ± 3.0	18.8 ± 0.6	21.2 ± 4.2	19.6 ± 4.1	0.0 ± 0.0	46.7 ± 3.3	32.7 ± 4.3	20.6 ± 2.4	0.0 ± 0.0

**Table 12 cells-14-01847-t012:** Tabulated summary of the relative frequency (%) of FB^+^/PNX^+^/PACAP^+^ cells in the L and the S DRG examined depending on their diameter. The data obtained were pooled in all the animals and presented as mean ± SEM, referring to the number of animals (*N* = 6). Asterisks mark statistically significant differences: ****** *p* = 0.0051, ****** *p* = 0.0051.

Segments of the Spinal Cord (%)					
L			S		
Small **	Medium **	Large	Small **	Medium **	Large
55.8 ± 2.2	44.2 ± 2.2	0.0 ± 0.0	83.6 ± 4.4	16.4 ± 4.4	0.0 ± 0.0

**Table 13 cells-14-01847-t013:** The relative frequency of FB^+^/PNX^+^/PACAP^+^ neurons in the particular intraganglionic domains of the L and the S DRG studied. Data obtained from all the animals studied were pooled and presented as mean ± SEM, referring to the number of animals (*N* = 6). Asterisks mark statistically significant differences: ***** *p* = 0.0217; ***** *p* = 0.0283; ******* *p* = 0.0009.

Segments of the Spinal Cord (%)									
L					S				
Cn ***	Cd	Cr	P *	Md *	Cn ***	Cd	Cr	P *	Md *
13.7 ± 1.5	23.3 ± 1.6	24.6 ± 1.1	26.6 ± 1.7	11.8 ± 3.5	0.0 ± 0.0	9.5 ± 9.5	37.3 ± 6.9	53.2 ± 7.0	0.0 ± 0.0

**Table 14 cells-14-01847-t014:** Tabulated summary of the relative frequency (%) of FB^+^/PNX^+^/GAL^+^ cells in the L and the S DRG examined depending on their diameter. Data obtained from all the animals studied were pooled and presented as mean ± SEM, referring to the number of animals (*N* = 6).

Segments of the Spinal Cord (%)					
L			S		
Small	Medium	Large	Small	Medium	Large
100.0 ± 0.0	0.0 ± 0.0	0.0 ± 0.0	100.0 ± 0.0	0.0 ± 0.0	0.0 ± 0.0

**Table 15 cells-14-01847-t015:** The relative frequency of FB^+^/PNX^+^/GAL^+^ neurons in the particular intraganglionic domains of the L and the S DRG studied. Data obtained from all the animals studied were pooled and presented as mean ± SEM, referring to the number of animals (*N* = 6).

Segments of the Spinal Cord (%)									
L					S				
Cn	Cd	Cr	P	Md	Cn	Cd	Cr	P	Md
3.7 ± 3.7	41.5 ± 5.9	37.4 ± 8.1	3.7 ± 3.7	13.7 ± 8.7	0.0 ± 0.0	33.3 ± 16.6	66.7 ± 16.6	0.0 ± 0.0	0.0 ± 0.0

**Table 16 cells-14-01847-t016:** Tabulated summary of the relative frequency (%) of FB^+^/PNX^+^/CRT^+^ cells in the L and the S DRGs examined depending on their diameter. Data obtained from all the animals studied were pooled and presented as mean ± SEM, referring to the number of animals (*N* = 6). Asterisks mark statistically significant differences: ****** *p* = 0.0022; ******* *p* = 0.0004.

Segments of the Spinal Cord (%)					
L			S		
Small ***	Medium **	Large	Small ***	Medium **	Large
61.3 ± 5.6	38.7 ± 5.6	0.0 ± 0.0	0.0 ± 0.0	0.0 ± 0.0	0.0 ± 0.0

**Table 17 cells-14-01847-t017:** The relative frequency of FB^+^/PNX^+^/CRT^+^ neurons in the particular intraganglionic domains of the L and the S DRGs studied. Data obtained from all the animals studied were pooled and presented as mean ± SEM, referring to the number of animals (N = 6).

Segments of the Spinal Cord (%)									
L					S				
Cn	Cd	Cr	P	Md	Cn	Cd	Cr	P	Md
0.0 ± 0.0	22.2 ± 11.1	27.8 ± 14.7	22.2 ± 22.2	27.8 ± 14.7	0.0 ± 0.0	0.0 ± 0.0	0.0 ± 0.0	0.0 ± 0.0	0.0 ± 0.0

**Table 18 cells-14-01847-t018:** Tabulated summary of the relative frequency (%) of FB^+^/PNX^+^/SOM^+^ cells in the L and the S DRGs examined depending on their diameter. Data obtained from all the animals studied were pooled and presented as mean ± SEM, referring to the number of animals (*N* = 6).

Segments of the Spinal Cord (%)					
L			S		
Small	Medium	Large	Small	Medium	Large
66.6 ± 21.1	33.0 ± 21.1	0.0 ± 0.0	66.6 ± 21.1	33.3 ± 21.1	0.0 ± 0.0

**Table 19 cells-14-01847-t019:** The relative frequency of FB^+^/PNX^+^/SOM^+^ neurons in the particular intraganglionic domains of the L and the S DRGs studied. Data obtained from all the animals studied were pooled and presented as mean ± SEM, referring to the number of animals (*N* = 6).

Segments of the Spinal Cord (%)									
L					S				
Cn	Cd	Cr	P	Md	Cn	Cd	Cr	P	Md
100 ± 0.0	0.0 ± 0.0	0.0 ± 0.0	0.0 ± 0.0	0.0 ± 0.0	0.0 ± 0.0	50.0 ± 21.8	50.0 ± 21.8	0.0 ± 0.0	0.0 ± 0.0

**Table 20 cells-14-01847-t020:** The relative numbers of individual, differently neurochemically coded subpopulations of PNX^+^ sensory neurons supplying the uterine cervix, as determined by the analysis of consecutive serial sections of the DRG studied.

Colocalization Patterns of PNX with Different Neurotransmitters in the Uterine Cervix-supplying DRG Neurons	%
FB^+^/PNX^+^	4.3 ± 0.7
FB^+^/PNX^+^/CGRP^+^	10.4 ± 4.1
FB^+^/PNX^+^/SP^+^	5.6 ± 0.8
FB^+^/PNX^+^/PACAP^+^	0.2 ± 0.2
FB^+^/PNX^+^/GAL^+^	0.2 ± 0.2
FB^+^/PNX^+^/CRT^+^	0.6 ± 0.6
FB^+^/PNX^+^/CGRP^+^/NOS^+^	5.1 ± 1.5
FB^+^/PNX^+^/CGRP^+^/PACAP^+^	1.5 ± 0.2
FB^+^/PNX^+^/CGRP^+^/GAL^+^	0.2 ± 0.2
FB^+^/PNX^+^/CGRP^+^/CRT^+^	0.3 ± 0.3
FB^+^/PNX^+^/CGRP^+^/SP^+^	21.3 ± 3.8
FB^+^/PNX^+^/SP^+^/PACAP^+^	0.3 ± 0.1
FB^+^/PNX^+^/SP^+^/NOS^+^	0.7 ± 0.3
FB^+^/PNX^+^/SP^+^/GAL^+^	0.9 ± 0.8
FB^+^/PNX^+^/SP^+^/CRT^+^	0.8 ± 0.5
FB^+^/PNX^+^/SP^+^/SOM^+^	0.3 ± 0.3
FB^+^/PNX^+^/CGRP^+^/SP^+^/PACAP^+^	11.3 ± 2.2
FB^+^/PNX^+^/CGRP^+^/SP^+^/CRT^+^	0.3 ± 0.3
FB^+^/PNX^+^/CGRP^+^/SP^+^/SOM^+^	0.3 ± 1.1
FB^+^/PNX^+^/CGRP^+^/SP^+^/GAL^+^	3.2 ± 1.2
FB^+^/PNX^+^/CGRP^+^/SP^+^/NOS^+^	19.1 ± 3.3
FB^+^/PNX^+^/PACAP^+^/GAL^+^	0.5 ± 0.0
FB^+^/PNX^+^/CGRP^+^/PACAP^+^/NOS^+^	1.0 ± 0.1
FB^+^/PNX^+^/SP^+^/PACAP^+^/NOS^+^	0.2 ± 0.2
FB^+^/PNX^+^/SP^+^/PACAP^+^/GAL^+^	0.8 ± 0.8
FB^+^/PNX^+^/CGRP^+^/SP^+^/PACAP^+^/NOS^+^	8.0 ± 0.8
FB^+^/PNX^+^/CGRP^+^/SP^+^/PACAP^+^/GAL^+^	0.4 ± 0.3
FB^+^/PNX^+^/CGRP^+^/SP^+^/NOS^+^/GAL^+^	0.3 ± 0.1
FB^+^/PNX^+^/CGRP^+^/SP^+^/CRT^+^/PACAP^+^	0.4 ± 0.4
FB^+^/PNX^+^/SP^+^/PACAP^+^/CRT^+^/NOS^+^/GAL^+^	0.2 ± 0.1
FB^+^/PNX^+^/SP^+^/CGRP^+^/CRT^+^/PACAP^+^/NOS^+^	0.4 ± 0.0
FB^+^/PNX^+^/SP^+^/CGRP^+^/CRT^+^/PACAP^+^/GAL^+^	0.9 ± 0.0

## Data Availability

The data that support the findings of this study are available from the corresponding author upon reasonable request.

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
