# Peer review of "Distribution and Neurochemical Characterization of Dorsal Root Ganglia (DRG) Neurons Containing Phoenixin (PNX) and Supplying the Porcine Uterine Cervix"

_cells, 2025, doi:10.3390/cells14231847_

Round 1

Reviewer 1 Report

Comments and Suggestions for Authors

This study, by using retrograde neuronal labelling with the fluorescent dye Fast blue, describes the segmental innervation pattern of the uterine cervix. Significant differences in the sensory innervation of the uterine cervix by lumbar and sacral DRG neurons were observed. The authors also discuss marked species differences in the segmental distribution of DRG neurons serving the uterine cervix.

Importantly, this quantitative morphometric immunohistochemical study describes distinct subpopulations of dorsal root ganglion neurons which express phoenixin and innervate the uterine cervix. Colocalizations of phoenixin with other sensory neuropeptides are also demonstrated. The possible functional significance of the colocalizations of phoenixin with CGRP, SP, PACAP, GAL, CRT, SOM and nNOS is also discussed.

Whereas the authors lay significant emphasis on the role of phoenixin-immunoreactive DRG neurons in pain mechanisms, the localization of TRP ion channels, such as TRPV1 and TRPA1, critical in the transmission of nociceptive impulses and local tissue reactions, has not been attempted. The authors should make a comment on this issue.

The discussion of the data is appropriate and thorough. It would be of some interest, however, if the authors would comment on the possible role of sensory nerves in tumor growth and progression in the uterine cervix which is frequently affected by malignant processes. In many organs sensory nerves have been shown to affect tumor pathology (cf. Zhi et al., Nature. 640:802-810. (2025), doi: 10.1038/s41586-025-08591-1), including the cervix uteri (e.g., Horn et al., J Cancer Res Clin Oncol, 136:1557-62 (2010) doi: 10.1007/s00432-010-0813-z).

Minor comments.

The demonstration of the intraganglionic distribution of labelled DRG neurons would be more illustrative by showing them as color-coded dots or areas on schematic drawings of the DRG.

Tables 4-21: Although the reader can figure out the data belonging to the L and S segments, respectively, the separation of the data e.g., by a vertical line would be helpful.

The use of spinal cord segments instead of neuromeres would probably more appropriate. Similarly, for example, DRG of thoracic neuromeres may be replaced with thoracic DRG.

Reviewer 2 Report

Comments and Suggestions for Authors

This study provided very detailed information about the adistribution and Neurochemical characterization of dorsal root ganglia neurons containing Phoenixin (PNX) and supplying porcine uterine cervix, although the pigs may not be very important like women. Overall, it is well designed, carefully conducted and analyzed. The manuscript was well organized and written in fluent English. A few minor issues need to be checked/improved. 

  1. Since the manuscript includes too many tables, the reviewer suggests the authors put Table 1 and Table 2 into the supplemental material.
  2. Please try to make the subtitles short. For example, "3.1. The Distribution Pattern and Morphometrical Characteristics of DRG Sensory Neu-rons supplying the Porcine Uterine Cervix" could be shortened as: Distribution Pattern and Morphometrical Characteristics of Porcine UC-SNs.
  3. In Fig 1, did the *** on the top of L and S indicate the differeces between these groups and Th groups (Did the Th groups serve as the control groups) ? Please make it clear.
  4. Is the FB staining a specifical staining method for neurons? What's those small blue dots in panel a and b of Fig 2?
  5. For Fig 7, do these Y-axises need more detailed labeling instead of "%"? 
